# Pd-Based Nano-Catalysts Promote Biomass Lignin Conversion into Value-Added Chemicals

**DOI:** 10.3390/ma16145198

**Published:** 2023-07-24

**Authors:** Ming Zhao, Liang Zhao, Xiao-Yan Zhao, Jing-Pei Cao, Koh-ichi Maruyama

**Affiliations:** 1Department of Materials and Biology, National Institute of Technology, Akita College, Akita 011-8511, Japan; maruko@akita-nct.ac.jp; 2Jiangsu Province Engineering Research Center of Fine Utilization of Carbon Resources, China University of Mining & Technology, Xuzhou 221116, China; zhaoxiaoyan@cumt.edu.cn (X.-Y.Z.); caojingpei@cumt.edu.cn (J.-P.C.); 3Institute of Innovative Research, Tokyo Institute of Technology, 4259 Nagatsuta, Midori-ku, Yokohama 226-8503, Japan

**Keywords:** palladium, phosphorus, lignin, selective hydrogenation, hydrolysis, guaiacol

## Abstract

Lignin, as a structurally complex biomaterial, offers a valuable resource for the production of aromatic chemicals; however, its selective conversion into desired products remains a challenging task. In this study, we prepared three types of Pd-based nano-catalysts and explored their application in the depolymerization of alkali lignin, under both H_2_-free (hydrogen transfer) conditions and H_2_ atmosphere conditions. The materials were well characterized with TEM, XRD, and XPS and others, and the electronic interactions among Pd, Ni, and P were analyzed. The results of lignin depolymerization experiments revealed that the ternary Pd-Ni-P catalyst exhibited remarkable performance and guaiacols could be produced under H_2_ atmosphere conditions in 14.2 wt.% yield with a selectivity of 89%. In contrast, Pd-Ni and Pd-P catalysts resulted in a dispersed product distribution. Considering the incorporation of P and the Pd-Ni synergistic effect in the Pd-Ni-P catalyst, a possible water-involved transformation route of lignin depolymerization was proposed. This work indicates that metal phosphides could be promising catalysts for the conversion of lignin and lignin-derived feedstocks into value-added chemicals.

## 1. Introduction

The rising consumption of fossil fuels has resulted in growing concerns over environmental pollution and the emerging energy shortage [1,2,3]. Biomass, as an abundant renewable resource, holds significant potential for the sustainable development of a future world [4,5]. Among the components of lignocellulose, lignin is the most stable and the only polymer consisting of abundant aromatic rings. Therefore, lignin is a valuable raw material for the production of aromatic chemicals, which could be further converted into jet fuels, pharmaceutical intermediates, and other organic polymers [6,7,8,9].

Catalytic depolymerization of lignin has emerged as one of the most promising strategies for lignin utilization [10]. In the past few decades, many types of transition metal catalysts, mainly groups VI, VIII, IX, and X, were used for lignin depolymerization [11,12]. In contrast to nickel-based catalysts, the platinum group metals (Pd, Pt, Ru, Rh, and Ir) exhibited higher intrinsic activity in hydrotreating reactions, including the hydrogenolysis of lignin [13,14,15,16]. Typically, Shu et al. reported an efficient lignin hydrogenolysis process using Pd/C catalysts in combination with metal chlorides, and the distribution of phenolic products could be regulated by varying the types of metal chlorides [17]. Bimetallic catalysts have also gained increasing attention due to the electronic and synergistic effects between two metals [4,18,19]. For instance, the combination of Zn and Pd/C was able to cleave the C-O ether bond of lignin under relatively mild reaction conditions (150 °C and 2 MPa H_2_), and then promote hydrodeoxygenation of the alcohol group, giving phenols as the depolymerized products selectively [20].

Recently, the doping of non-metallic elements into metal-based catalysts has been explored to enhance catalytic activity and product selectivity for lignin depolymerization, ascribed to nanostructure modification and the coordination effect of the non-metallic atoms. Li et al. synthesized Ru immobilized on N-doped graphene carbon catalysts (Ru@N-doped carbon) with the Ru nanoparticles stabilized by N atoms and the defect-rich carbon structure. Compared with Pd/C and Ru/C, the Ru@N-doped carbon catalyst contributed to higher catalytic activity for lignin depolymerization [21]. Chen et al. found that the addition of phosphorus increased the presence of PO_3_^−^ and PO_4_^3−^ sites in the 5P-Mo/SEP catalyst. These sites could act as Brønsted acids to promote the activation of the solvent ethanol to serve as an in situ hydrogen source, which consequently facilitates the hydrogenation of reactive intermediates leading to the formation of phenolic compounds, particularly guaiacol [22].

Despite the rapid development of the catalytic conversion of lignin, directing the reaction pathways towards desired chemicals remains a major challenge due to uncontrolled interactions between the catalyst active sites and lignin molecules. The selectivity of the desired product has always been unsatisfactory. In our previous study, it was demonstrated that the addition of P in Pd-based catalysts could make Pd positively charged to reduce the stability of Pd-H [22], leading to high selectivity for the partial reduction and the reductive hydrolysis of lignin derivatives [23,24,25]. Herein, we aim to further study the catalysis of metal phosphides in the depolymerization of a real lignin, alkali lignin. Four types of Pd-based catalysts including Pd, Pd-P, Pd-Ni, and Pd-Ni-P were prepared and evaluated for the catalytic conversion of alkali lignin. It was suggested that P and Ni atoms in the Pd-Ni-P catalyst could play synergistic effects with Pd to adsorb -OH in lignin molecules and dissociate H_2_O, respectively, which led to the selective hydrolysis of lignin. Pd-Ni-P represents a totally novel type of Pd-based catalyst for biomass polymer degradation through tandem reactions. Pd-Ni-P exhibited an exceptional selectivity of 89% in producing guaiacols in a high yield of 14.1 wt.%.

## 2. Materials and Methods

Information on the chemical reagents used in this work is provided in the Appendix A. Alkali lignin was purchased from Sigma-Aldrich; the element composition is shown in Table 1.

### 2.1. Preparation of Materials

Pd/C: The catalyst was purchased from Beijing Hawk Science & Technology Co., Ltd. (Beijing, China). It contained 10 wt.% of Pd metal.

Pd-Ni/C: The catalyst was synthesized by an impregnation method. Specifically, Pd(acac)_2_ (0.1 g) and Ni(acac)_2_ (0.09 g) were dissolved in 20 mL dichloromethane; after stirring for 30 min, 0.3 g activated carbon was added into the aqueous solution. The mixed solution was maintained in a vacuum oven for 24 h. Then, the mixture was calcined under an Ar flow of 60 mL/min at 500 °C for 2 h, followed by reduction at the same flow rate with H_2_ gas at 500 °C for 2 h.

Pd-Ni-P/C: Pd(acac)_2_ (0.2 mmol), Ni(acac)_2_ (0.2 mmol), triphenylphosphine (TPP, 0.88 mmol), tetrabutylammonium bromide (TBAB, 1.5 mmol), trioctylphosphine oxide (TOPO, 3 mmol), and oleylamine (OLA, 6 mL) were mixed in an autoclave reactor, and N_2_ was poured three times into the reactor to remove the air inside. The reaction solution was initially stirring for 15 min without heating, and then heated to 220 °C and maintained for 30 min. Then, the reaction solution was cooled to room temperature and mixed with 30 mL ethanol for washing and centrifugation. The centrifugal speed was 11,000 rpm and the centrifugation time was 10 min. The precipitate was collected and washed repeatedly with ethanol to obtain Pd-Ni-P NPs; then, the activated carbon (weight ratio of NPs: support = 3:7) in hexane was sonicated for 30 min, and a suspension of Pd-Ni-P NPs in hexane was added. The resultant mixture was sonicated again for a further 30 min to obtain a homogeneous solution. After evaporating the hexane, the initial Pd-Ni-P/C was obtained. The final Pd-Ni-P/C was obtained after annealing treatment under 500 °C in a N_2_ atmosphere for 2 h [23].

Pd-P/C: The catalyst was prepared in the same way as Pd-Ni-P/C in the absence of Ni(acac)_2_.

### 2.2. Catalytic Depolymerization of Lignin

The hydrogenation depolymerization of lignin was conducted in a high-pressure reactor with a volume of 100 mL. A typical reaction proceeded as follows: 30 mg of catalyst, 100 mg of lignin, and 20 mL of solvent were added to the reactor. To eliminate air from the reactor, H_2_/Ar gas was introduced and discharged three times, resulting in an initial H_2_/Ar pressure of 2 MPa. The reactor was then heated to 250 °C and maintained at this temperature for approximately 25 min. Stirring was carried out at 800 rpm for 4 h.

The identification and quantification of the depolymerization products were performed using GC-MS analysis. The employed instrument was the Trace1300-ISQ7000 gas spectrometer manufactured by Thermo Fisher Scientific Inc. located in Waltham, MA, United States. The type and relative content of the depolymerization products were determined by constructing product standard curves, as indicated by the following Equations (1)–(3). The concentrations of guaiacols and phenols were calculated using Equation (1) with the guaiacol standard curve. The concentrations of long-chain oxygenate products were calculated using Equation (2) with the n-butanol standard curve. Additionally, the concentrations of other products (benzenes) were determined using Equation (3) with the ethylbenzene standard curve. In these equations, A represents the relative peak area, and C represents the product concentration.

The product yield was determined based on the original amount of lignin employed, while the product selectivity was calculated based on the total products, as shown in Equations (4) and (5).
(1)$$A_{\mathrm{guaiacols}/\mathrm{phenols}}=1.47\times 10^{10}\times C_{\mathrm{guaiacols}/\mathrm{phenols}}+1.0\times 10^{6}\;$$
(2)$$A_{\mathrm{long}-\mathrm{chain}\;\mathrm{oxygenates}}=3.24\times 10^{8}\times C_{\mathrm{long}-\mathrm{chain}\;\mathrm{oxygenates}}+2.1\times 10^{5}$$
(3)$$A_{\mathrm{benzenes}}=1.01\times 10^{10}\times C_{\mathrm{benzenes}}+1.54\times 10^{4}\;$$
(4)$${\mathrm{Yield}}_{\mathrm{product}}=\frac{\mathrm{mass}\;\mathrm{of}\;\mathrm{product}}{\mathrm{mass}\;\mathrm{of}\;\mathrm{original}\;\mathrm{lignin}}=\frac{C_{\mathrm{product}}\times V_{\mathrm{total}\;\mathrm{liquid}\;\mathrm{products}}}{\mathrm{mass}\;\mathrm{of}\;\mathrm{original}\;\mathrm{lignin}}$$
(5)$${\mathrm{Selectivity}}_{\mathrm{product}}=\frac{\mathrm{mass}\;\mathrm{of}\;\mathrm{product}}{\mathrm{mass}\;\mathrm{of}\;\mathrm{all}\;\mathrm{products}}=\frac{C_{\mathrm{product}}\times V_{\mathrm{total}\;\mathrm{liquid}\;\mathrm{products}}}{\mathrm{mass}\;\mathrm{of}\;\mathrm{all}\;\mathrm{products}}$$

### 2.3. Characterizations

X-ray diffraction (XRD) was performed to test the crystalline structures of the samples using a Bruker D8 Advance diffractometer supplied by Bruker Corporation in Karlsruhe, Germany with Cu-Kα radiation at 40 kV and 40 mA. X-ray photoelectron spectroscopy (XPS) was performed using a ESCALAB 250Xi X-ray photoelectron spectrometer supplied by Thermo Fisher Scientific Inc. with Al-Kα as the photon source. Transmission electron microscopy (TEM) was performed using a JEOL JEM-2100 (HR) instrument supplied by JEOL Ltd. Located in Tokyo, Japan operated at 200 kV to analyze the morphology of the samples.

The porous structure of the catalysts was measured with a V-Sorb 4800TP N2 adsorption-desorption instrument supplied by Beijing CI-Ultrametrics Technology Co., Ltd. in Beijing, China. The catalysts were degassed under a N_2_ atmosphere at 300 °C to remove the gas and moisture adsorbed on the catalysts before the tests. The total pore volume (TPV), the specific surface area (SSA), and the average pore diameter (Da) were analyzed according to the adsorbed volume of nitrogen at a relative pressure P/P0 of 0.99, the multipoint BET equation, and the BJH method, respectively. The measurement error of the N_2_ adsorption-desorption instrument was less than 1.5%.

## 3. Results

### 3.1. Catalyst Preparation and Characterizations

Three types of Pd-based catalysts (Pd-P/C, Pd-Ni/C, and Pd-Ni-P/C) were prepared based on our previous work (see Section 2. Materials and Methods for the details) [24]. Transmission electron microscopy (TEM) indicated the particle sizes of Pd-P, Pd-Ni, and Pd-Ni-P were 11 nm, 7.1 nm, and 8.1 nm, respectively (Figure 1). The commercial Pd/C showed a similar NP size of about 10 nm (Appendix A). Although the average particle sizes were similar between those of Pd-P and Pd-Ni-P, the TEM images showed that there were large particles in the Pd-P samples. As the Pd-P/C and Pd-Ni-P/C catalysts were prepared by annealing under 500 °C, adding Ni could probably stabilize the thermostability of NPs from aggregation. Based on our previous study, the sizes of the Pd-Ni-P NPs increased gradually from 7.0 nm to 8.1 nm when they were calcined from 300 °C to 500 °C [24]. This might imply that the atoms mainly diffused within the particle rather than suffering Ostwald ripening. In comparison, Pd-P might be less thermostable, with parts of the NPs suffering Ostwald ripening to form large ones. The Brunauer–Emmett–Teller (BET) surface areas of Pd-P/C, Pd-Ni/C, and Pd-Ni-P/C were determined as 90 m^2^/g, 96 m^2^/g, and 134 m^2^/g, respectively (Appendix A). Pd-Ni-P/C exhibited a larger area than Pd-P/C due to its smaller particle size. On the other hand, the surface area of Pd-Ni/C did not correspond closely to its particle size, probably due to the different synthesis methods used. The four types of Pd-based materials contained similar Pd contents (7.3–10.4 wt.%, Appendix A) as determined by inductively coupled plasma-mass spectrometry (ICP-MS). The atomic compositions of Pd-P/C, Pd-Ni/C, and Pd-Ni-P/C were calculated to be Pd_72_P_28_/C, Pd_36_Ni_64_/C, and Pd_34_Ni_39_P_27_/C, respectively, based on the analysis results of energy dispersive X-ray spectroscopy (EDS) and ICP-MS. 

X-ray diffraction (XRD) was performed to examine the crystalline structures of Pd-Ni/C, Pd-P/C, and Pd-Ni-P/C (Figure 2). Compared to the face-centered cubic (fcc) Pd (PDF#05-0681-Pd), all the signals of Pd-Ni/C shifted to larger angles due to the addition of Ni atoms. For Pd-P/C, the fcc peaks shifted a little to smaller angles because of the insertion of P atoms into the Pd phase. The XRD pattern of Pd-Ni-P/C closely matched that of fcc-Pd, probably due to an integrated effect of Ni and P doping. This also suggested that ternary alloy formed in the Pd-Ni-P NPs. All three materials showed three major peaks at 111, 200, and 220, indicating the face-centered cubic (fcc) structure was the dominant structure among them. 

To study the electronic interactions between different atoms in the Pd-based NPs, X-ray photoelectron spectroscopy (XPS) was performed (Figure 2b–d). In Figure 2b, the binding energies (BEs) of Pd3d_5/2_ in Pd-Ni/C, Pd-P/C, and Pd-Ni-P/C were observed to be 336.0 eV, 336.0 eV, and 335.7 eV, respectively. When the Pd signals of the Pd-Ni-P/C and Pd-P/C catalysts were deconvoluted (Appendix A), it was observed that metallic Pd was the main component and some Pd oxide occurred. Compared with the BE of Pd3d_5/2_ in metallic Pd(0) (335.0 eV) reported in ref. [23], positive shifts in Pd-Ni/C (336.0 eV), Pd-P/C (336.0 eV), and Pd-Ni-P/C (335.7 eV) were observed. This indicated that the electron flows occurred from Pd to Ni in Pd-Ni/C and from Pd to P in Pd-P/C. The BEs of Pd3d in Pd-Ni-P were slightly lower than those in Pd-Ni and Pd-P, probably due to the integrated electronic interactions among Pd, Ni, and P where electron transfer from Ni to P also occurred. This transfer can also be observed in Figure 2c and Appendix A, in which the peaks of NiO and Ni(OH)_2_ can be seen to shift into larger BEs in the case of Pd-Ni-P/C compared to those in Pd-Ni/C. This might indicate that partial electrons in Ni were robbed by O and P atoms at the same time. Moreover, there was a clear signal of Ni metal for Pd-Ni/C (Appendix A). It is suggested that the catalytic property of metallic Ni plays a role in this catalyst. In Figure 2d, the main peak observed is assigned to P-O and there is a small signal of P(0) in Pd-P/C.

Oxidized states of Ni were detected in the XPS spectra for both the Pd-Ni and Pd-Ni-P samples; however, no NiO peaks (37.3°, 43.3°, 62.9°, and 87.0° of 2θ) were observed in the XRD patterns (Figure 2a). This might also indicate the formation of Pd-Ni alloy and Pd-Ni-P alloy.

### 3.2. H_2_-Free Lignin Depolymerization

To investigate the possibility of H_2_-free lignin depolymerization, the catalytic reactions were initially conducted under an Ar atmosphere with isopropanol as the solvent at 250 °C. In this case, lignin itself and the proton solvent served as the hydrogen sources. The obtained liquid product was analyzed by GC-MS (Figure 3a and Table 2). Among the four catalysts, Pd/C exhibited the highest catalytic activity, resulting in liquid products in 19.2 wt.% yield. However, the selectivity of the guaiacol products was as low as 23% (Table 2 and Figure 3b). Figure 3a shows the color of the liquid product, indicating that Pd/C effectively catalyzed the conversion of a greater number of lignin molecules dissolved in the solvent, yielding a clear and bright lignin oil. On the other hand, the bimetallic Pd-Ni/C catalyst demonstrated the lowest activity, with a liquid product yield of 7.7 wt.%. Pd-Ni-P/C achieved a product yield of 13.5 wt.% under the Ar atmosphere, and the selectivity for guaiacols was 28%, which was lower than that of Pd-P/C. These Pd-based catalysts still facilitated the depolymerization of lignin to liquid products under an Ar atmosphere, primarily due to the cracking and breaking of lignin bonds at high temperatures. Moreover, the presence of the isopropanol solvent favored the provision of a hydrogen source [24]. However, wide distributions of the depolymerization products were observed for all four catalysts, indicating that the selectivity of the Pd-based catalysts was unsatisfactory in the hydrogen-transfer hydrotreatment of lignin.

### 3.3. Lignin Depolymerization in H_2_ Atmosphere

Next, the catalytic depolymerization of lignin was performed under a H_2_ atmosphere with an initial pressure of 2 MPa. The results are summarized in Figure 3a,c,d and Table 2. When using Pd/C as the catalyst, lignin was converted into liquid products with a yield of 13.2 wt.%. However, a broad product distribution, including guaiacol, phenols, and others, was observed with selectivities of 33%, 37%, and 30%, respectively. In the case of Pd-P/C, the selectivity for phenolic products was the highest, reaching 67%, with a corresponding yield of 9.6 wt.%. The bimetallic Pd-Ni/C catalyst displayed the lowest activity, resulting in a liquid yield of only 10.1 wt.%. Notably, Pd-Ni-P/C demonstrated a significant ability to produce guaiacols. The obtained liquid was the most colorless among those obtained (Figure 3a). Pd-Ni-P/C achieved a yield of 10.1 wt.%, with a remarkable selectivity of 82% for guaiacols.

### 3.4. Lignin Depolymerization in Different Solvents

The solvents used are recognized to play a crucial role in lignin conversion as their polarities can influence lignin dissolution and, consequently, its interaction with the catalyst surface [26]. In recent literature reports, combinations of organic solvents and water have been widely employed for the efficient degradation of lignin and its derivatives [27]. Herein, Figure 4 illustrates the results of lignin conversion catalyzed by Pd-Ni-P/C in different solvents in the absence/presence of water. Generally, the addition of water improved the selectivity towards guaiacols compared to a single-component solvent. Among the solvents tested, methanol exhibited the highest catalytic activity, leading to an increased yield of depolymerization products, with a total yield of 19.5 wt.%. This enhancement can be attributed to the release of hydrogen through the steam reforming of methanol, which subsequently participates in the catalytic transfer hydrogenation of lignin intermediates [28]. Similar results were also observed in the ethanol system. However, in both methanol and ethanol solvents, the yield and selectivity of guaiacols were relatively low due to the excessively strong hydrogenation ability, which compromised the integrity of the benzene ring in lignin and the intermediates. In contrast, when isopropanol was used as the solvent, the total product yield was 12.4 wt.%, with a guaiacol yield and selectivity of 10.1 wt.% and 82%, respectively. Isopropanol promotes the generation of water molecules through condensation, catalyzed by Pd catalysts [24]. Consequently, additional hydrolysis processes probably occurred within the lignin molecule, resulting in the increased production of guaiacol. Therefore, the addition of water to the solvent appears to be an effective approach to enhance the selectivity towards guaiacol products. As shown in Figure 4a,b, the selectivity of guaiacols was significantly improved in solvents containing water. In the mixed solvents of methanol/water and ethanol/water, the selectivities of guaiacol increased to 38% and 85%, respectively. In isopropanol/water, the highest guaiacol selectivity was achieved as 89%. However, the conversion of lignin depolymerization in pure water was quite low, yielding only 0.95 wt.% of liquid products, probably due to the poor dispersion of lignin and the catalyst in water. Furthermore, non-polar solvents, such as hexane, were found to be unfavorable for the depolymerization of lignin, with a conversion of less than 2.0 wt.%, despite an increase in guaiacol selectivity after the addition of water. These results are consistent with previous reports where the polarity of the solvent has been shown to play a crucial role in completing the swelling of the lignin structure, facilitating better contact between the solvent and lignin [28,29]. Therefore, the selection of a suitable solvent or combination of solvents is essential for the depolymerization of lignin and achievement of desired product outcomes.

## 4. Discussion

As one major challenge in the direct transformation of lignin stems from its structural complexity, a comprehensive understanding of the molecule structure is crucial for achieving high-value-added products. Typically, lignin is primarily composed of phenylpropyl groups and exhibits three main aromatic structures: syringyl (S), guaiacyl (G), and hydroxyphenyl (H) units [30]. As shown in Figure 5a, each unit is derived from one monophenolic alcohol: sinapyl alcohol, coniferyl alcohol, or p-coumaryl alcohol, respectively. During the depolymerization process, the low-energy C-O bonds (e.g., *β*-O-4) are first cleaved, leading to the formation of lignin fragments. These fragments can undergo various depolymerization pathways under different catalytic conditions. In Figure 5b, a catalyst-free blank experiment was performed under the standard conditions. It can be seen that the depolymerization products were broadly dispersed, indicating the catalyst is essential for high selectivity of the target products.

Figure 5c illustrates a lignin fragment composed of sinapyl alcohol and coniferyl alcohol, which represents a possible initial *β*-O-4 product of depolymerization as determined by GC-MS analysis. Our previous studies on Pd-P and Pd-Ni-P catalysts have demonstrated that the electron-deficient Pd species can selectively hydrolyze aromatic ethers [24,31]. In ternary Pd-Ni-P catalyst systems, the existence of Ni enhances the water dissociation ability while simultaneously inhibiting the hydrodeoxygenation process of guaiacols [32]. Therefore, in the presence of water, Pd-P/C and Pd-Ni-P/C could promote the reductive depolymerization of lignin fragments generating alkenes ([D] in Figure 5c) [33]. Herein, it is suggested that P could form coordinating bonds with the -OH groups of lignin molecules and, together with Pd, facilitate the hydrolysis of the *β*-O-4 ether bond. In other words, the P in the Pd-P or Pd-Ni-P catalyst might play a role in the adsorption of the -OH group of lignin, and, from this viewpoint, the two catalysts might have low acidity. The hydrogenation of [D] affords [F]. Consequently, [F] could undergo further dealkylation processes to give guaiacols [G] as the final products. On the other hand, Pd/C could also facilitate the stepwise cleavage of *β*-O-4 through the action of hydrogen, resulting in the final production of phenols [E].


## 5. Conclusions

In conclusion, the Pd-Ni-P catalyst exhibited exceptional performance in the catalytic conversion of alkali lignin. The experimental findings demonstrated that an isopropanol solvent was favorable for lignin conversion, and the addition of a small amount of water enhanced the production of guaiacols. Under the reaction conditions of 250 °C and 2 MPa H_2_ pressure, a Pd-Ni-P/C catalyst achieved a guaiacol yield of 14.1 wt.%, with an impressive selectivity of 89%. For comparison, the product information for lignin depolymerization for various previously reported metal-based catalysts was illustrated in Appendix A. Based on the control experiments, it was suggested that P and Ni atoms in the Pd-Ni-P catalyst could have synergistic effects with Pd to adsorb -OH in the lignin molecule and dissociate H_2_O, respectively, leading to the selective hydrolysis of lignin. Pd-Ni-P represents a totally novel type of Pd-based catalyst for biomass polymer degradation through tandem reactions. This work presents a promising strategy for the production of valuable oxygenated fine chemicals from lignin and highlights the remarkable selective catalysis of metal phosphides.

## Figures and Tables

**Figure 1 materials-16-05198-f001:**
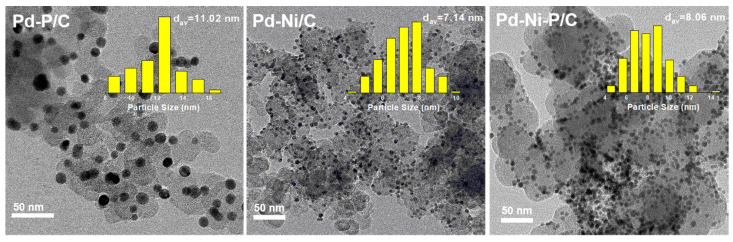
TEM images and particle size distribution of different Pd-based catalysts.

**Figure 2 materials-16-05198-f002:**
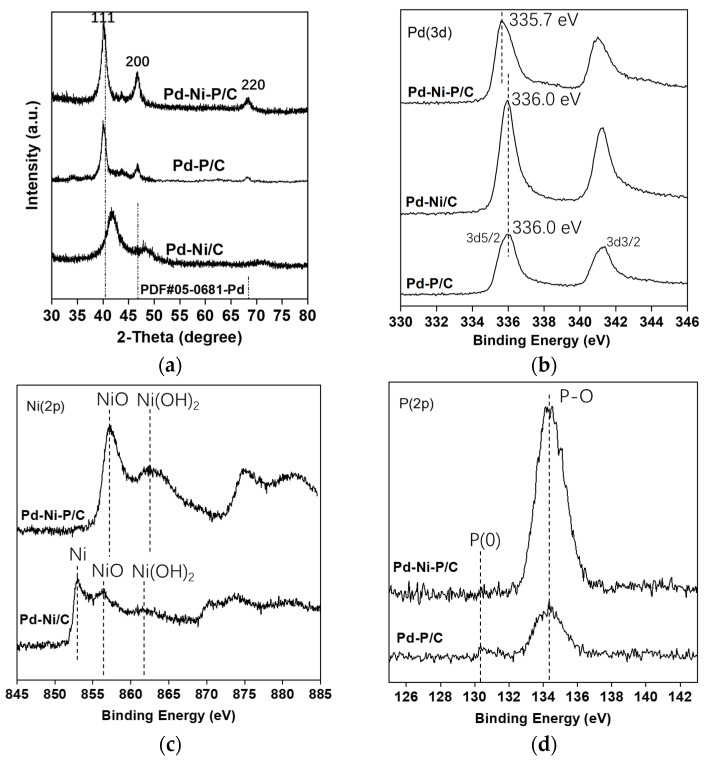
XRD patterns (**a**) and XPS spectra (**b**–**d**) of Pd-based catalysts.

**Figure 3 materials-16-05198-f003:**
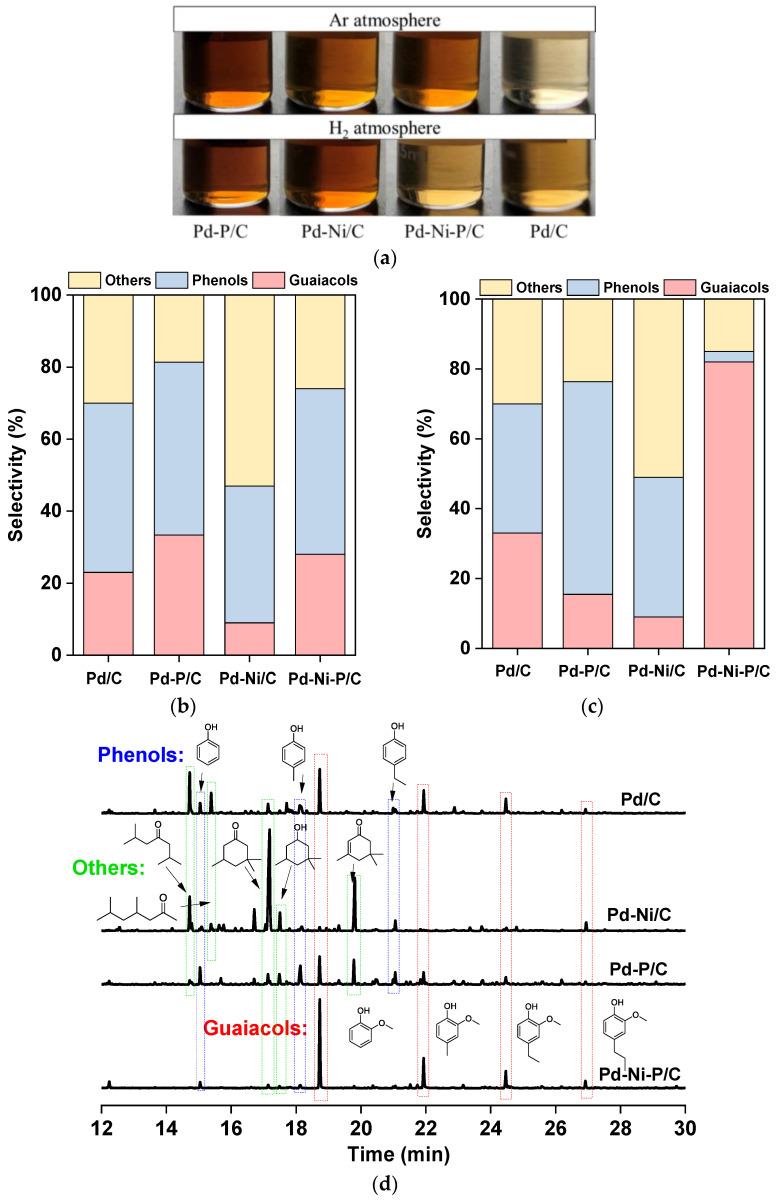
The obtained liquid of lignin depolymerization over different catalysts (**a**); product selectivity under Ar atmosphere (**b**), and H_2_ atmosphere (**c**); GC-MS spectra of the liquid products from lignin depolymerization (**d**). The product selectivity in (**b**,**c**) was determined based on the standard curves produced by guaiacol, ethylbenzene, and butanol with GC-MS.

**Figure 4 materials-16-05198-f004:**
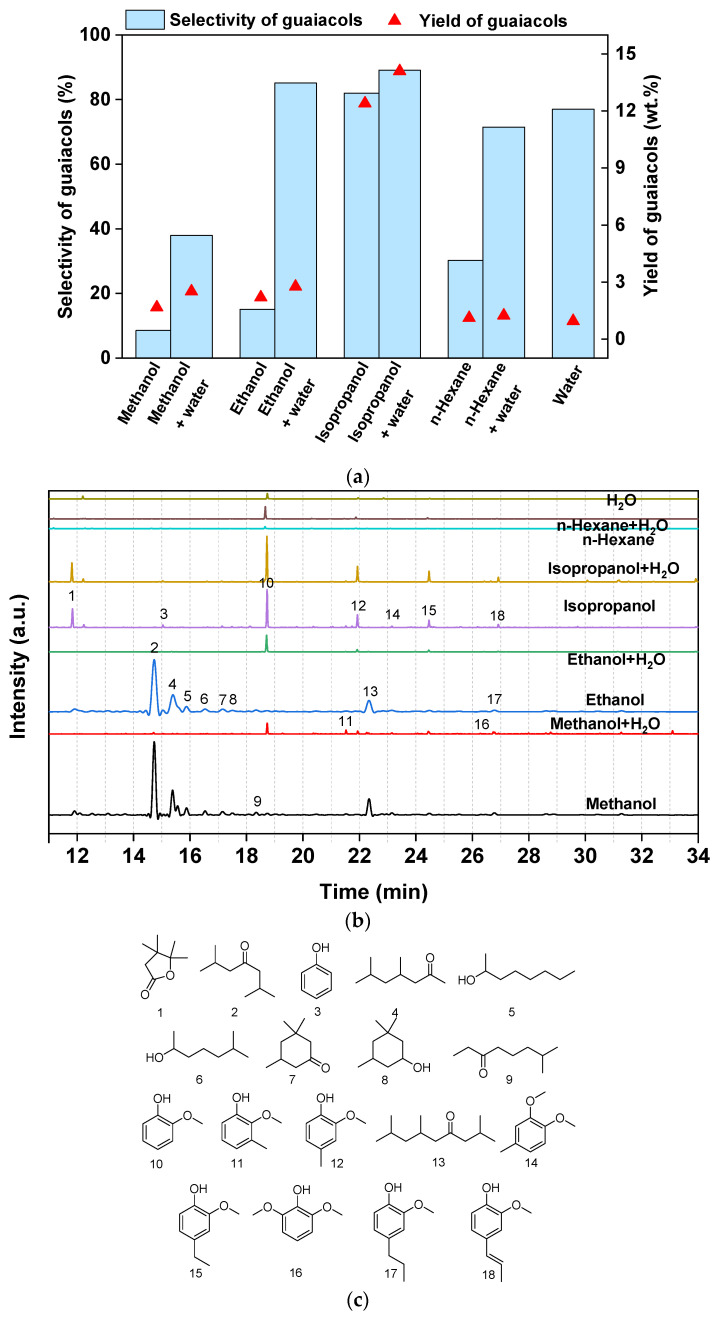
Product distribution of lignin depolymerization over Pd-Ni-P/C catalyst. (**a**) Yield and selectivity of guaiacols in different solvents (20 mL each) in the absence/presence of water (1 mL). (**b**,**c**) GC-MS analysis and list of main products of lignin depolymerization.

**Figure 5 materials-16-05198-f005:**
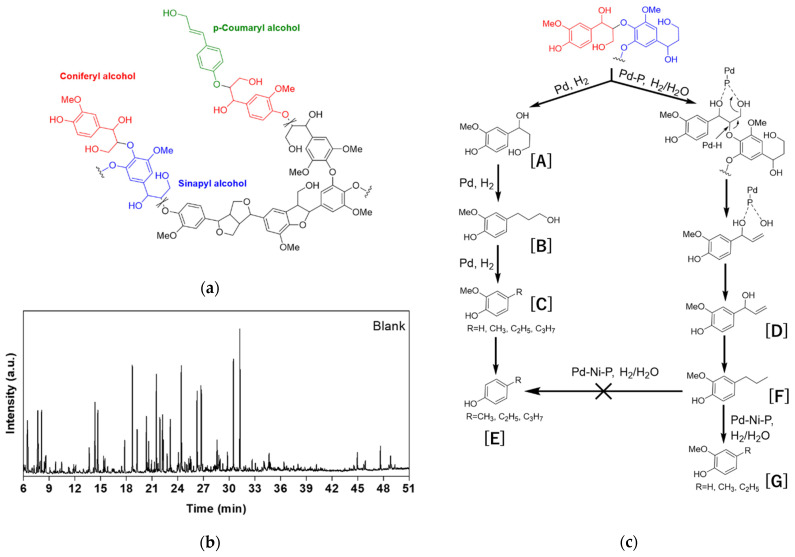
(**a**) Representative structure of lignin; (**b**) GC-MS spectra of lignin depolymerization products in the absence of catalyst. Reaction conditions: lignin (100 mg), isopropanol (20 mL), H_2_ (2 MPa), reaction temperature: 250 °C, reaction time: 4 h; (**c**) Possible route for lignin conversion over Pd-based catalysts. The reaction intermediates in sub-figure (**c**) are listed as follows. [A]: the hydrogenolysis product of lignin fragment, [B]: the dehydroxylation product of [A], [C]: guaiacols formed by the dihydroxylation or dealkylation product of [B], [E]: phenols formed by the demethoxylation of [C], [D]: allyl alcohol obtained by the reductive depolymerization of lignin fragment, [F]: the dehydroxylation/hydrogenation product of [D], and [G]: guaiacols formed by the dealkylation of [F].

**Table 1 materials-16-05198-t001:** Element composition of alkali lignin.

Composition (wt.%, daf ^1^)				
C	O ^2^	H	N	S ^3^
69.12	23.37	4.69	0.31	2.51

^1^ dry ash-free basis; ^2^ subtraction method; ^3^ total sulfur of desiccant.

**Table 2 materials-16-05198-t002:** Product distributions of lignin depolymerization under an Ar or a H_2_ atmosphere ^1^.

Catalyst	Atmosphere	Product Yields (wt.%)			
		Guaiacols	Phenols	Others	Total
Pd/C	Ar	4.4	9.1	5.7	19.2
Pd-P/C		3.3	4.8	1.8	9.7
Pd-Ni/C		0.7	2.9	4.1	7.7
Pd-Ni-P/C		3.8	6.2	3.5	13.5
Pd/C	H_2_	4.4	4.8	3.9	13.2
Pd-P/C		2.7	9.6	4.0	16.3
Pd-Ni/C		1.0	4.0	5.1	10.1
Pd-Ni-P/C		10.1	0.4	1.9	12.4

^1^ Reaction condition: lignin (100 mg), catalyst (50 mg), isopropanol (20 mL), Ar/H_2_ (2 MPa), reaction temperature: 250 °C, reaction time: 4 h. The product yields were calculated based on the standard curves produced by guaiacol, ethylbenzene, and butanol with GC-MS.

## Data Availability

The data presented in this study are available on request from the corresponding authors, mingzhao@akita-nct.ac.jp (M.Z.), and zhao.l.ae@m.titech.ac.jp (L.Z.). Data are contained within the article or the Appendix A.

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
