# Peer review of "Pd-Based Nano-Catalysts Promote Biomass Lignin Conversion into Value-Added Chemicals"

_materials, 2023, doi:10.3390/ma16145198_

Round 1

Reviewer 1 Report

This work studies the depolymerization of lignin as a biomass model compound for obtaining valuable molecules. The topic of the work is very interesting and current. The yields achieved towards the desired products are low, but in some cases, very high selectivity is reached, which makes the work attractive. However, before the publication of the work, I suggest that substantial changes be made to the manuscript. Mainly, the explanation of the results of the characterization techniques must be improved, and, then, correlate the catalytic results with the physicochemical properties of the catalytic materials with greater support.

- The catalytic results obtained in this work should be compared with others reported in the bibliography.

-The novelty of this work should be emphasized both in the introduction and in the conclusions.

-Line 65

“Pd-H [22].  leading”: Please check the text

-Line 140

“Transmission electron microscope (TEM) indicates the particle sizes of Pd-P, Pd-Ni, and 140 Pd-Ni-P were 11 nm, 7.1 nm, and 8.1 nm, respectively (Figure 1 and Figure S1).”

 Please discuss this result in more depth. Although the average size may be similar, the TEM images show that there are large particles in the Pd-P samples. Explain in more detail the effect of adding nickel on the particle size distribution. Figure S1 should be moved to the article.

-Why is there a greater dispersion of the metallic phases in the catalysts that contain Nickel?

What is the area of the catalysts? Please inform this, catalysis happens on the surface of the materials.

-Line 144:

"The compositions of Pd-P/C, Pd-Ni/C, and Pd-Ni-P/C were determined as Pd72P28/C, Pd36Ni64/C, and Pd34Ni39 P27/C, respectively". Are the percentages atomic?

-In catalysts containing nickel, this element exists in its oxidized state (as observed from the XPS results). In the XRD discussion, this result was not taken into account. The NiO diffraction lines appear at 37.3, 43.3, 62.9◦, and 87.0◦ of 2θ. Please review the discussion.

- The analysis of the XPS results needs to be improved. In the preparation of the catalysts, the authors do not indicate that the Pd-Ni-P/ catalyst is subjected to reduction treatment. The Pd-Ni-P/C and Pd-P/C catalysts were heat treated in inert (N2) up to 500°C.

 The Pd signal must be deconvoluted and the contributions of the oxidized and reduced species must be shown. This same analysis must also be carried out on the spectra of the catalysts for nickel.

-Despite the low yields, the catalytic results are very interesting. However, they should be discussed again after knowing more precisely the nature of the supported phases.

Could acidity provide an activity-promoting effect?

Did the authors study the reusability of the more selective catalyst?

Author Response

Dear reviewer,

We appreciate your constructive questions and comments. All your questions and comments were addressed, and the revised parts were highlighted with red color in the manuscript and supporting materials.

This work studies the depolymerization of lignin as a biomass model compound for obtaining valuable molecules. The topic of the work is very interesting and current. The yields achieved towards the desired products are low, but in some cases, very high selectivity is reached, which makes the work attractive. However, before the publication of the work, I suggest that substantial changes be made to the manuscript. Mainly, the explanation of the results of the characterization techniques must be improved, and, then, correlate the catalytic results with the physicochemical properties of the catalytic materials with greater support.

 - The catalytic results obtained in this work should be compared with others reported in the bibliography.

Answer 1. Thank you for your constructive suggestion. For comparison, the product information of lignin depolymerization promoted by various previously reported metal-based catalysts was illustrated in Figure S3. Please also refer to related statement the section of “5. Conclusions” on page 10. (Line 327)

-The novelty of this work should be emphasized both in the introduction and in the conclusions.

 Answer 2. Thank you for your constructive suggestion. The novelty of this work has been emphasized both in the introduction and in the conclusions. Please refer to the last paragraph of the Introduction part (Lines 61-71, page 2) and the conclusions part (Lines 329-333, pages 10-11).

-Line 65

“Pd-H [22].  leading”: Please check the text

Answer 3. Thank you for your correction. It has been revised. (Line 66)

-Line 140

“Transmission electron microscope (TEM) indicates the particle sizes of Pd-P, Pd-Ni, and Pd-Ni-P were 11 nm, 7.1 nm, and 8.1 nm, respectively (Figure 1 and Figure S1).”

Please discuss this result in more depth. Although the average size may be similar, the TEM images show that there are large particles in the Pd-P samples. Explain in more detail the effect of adding nickel on the particle size distribution. Figure S1 should be moved to the article.

Answer 4. Thank you for your constructive suggestion. The Pd-P/C and Pd-Ni-P/C catalysts were prepared by annealing under 500 oC. In our opinion, adding nickel could probably stabilize the thermostability of NPs from aggregation. Based on our previous study, the sizes of Pd-Ni-P NPs increased gradually from 7.0 nm to 8.1 nm when they were calcined from 300 oC to 500 oC (Chemical Engineering Journal 435 (2022) 134911). It might imply the atoms mainly diffuse within the particle rather than suffering Ostwald ripening. In comparison, Pd-P might be less thermostable and parts of the NPs suffered Ostwald ripening to form large ones. Please refer to the section “3.1. Catalyst Preparation and Characterizations” for the revisions. (Lines 158-166)

Figure S1 have been moved to the article as Figure 1.

-Why is there a greater dispersion of the metallic phases in the catalysts that contain Nickel?

What is the area of the catalysts? Please inform this, catalysis happens on the surface of the materials.

Answer 5. Thank you for your good question. The Pd-P/C and Pd-Ni-P/C catalysts were prepared by annealing under 500 oC. In our opinion, adding Ni could probably stabilize the thermostability of NPs from aggregation; while Pd-P might be less thermostable and parts of the NPs suffered Ostwald ripening to form large ones. The results of Brunauer-Emmett-Teller (BET) surface area also indicate the Pd-P/C have a lowest area (90 m2/g), compared to Pd-Ni/C (96 m2/g) and Pd-Ni-P/C (134 m2/g). (Lines 166-170)

-Line 144:

"The compositions of Pd-P/C, Pd-Ni/C, and Pd-Ni-P/C were determined as Pd72P28/C, Pd36Ni64/C, and Pd34Ni39 P27/C, respectively". Are the percentages atomic?

Answer 6. Yes, it is atomic percentages and this sentence was revised accordingly in the revised manuscript.

-In catalysts containing nickel, this element exists in its oxidized state (as observed from the XPS results). In the XRD discussion, this result was not taken into account. The NiO diffraction lines appear at 37.3, 43.3, 62.9◦, and 87.0◦ of 2θ. Please review the discussion.

Answer 7. Thank you for your constructive comment. In the revised manuscript, we added the related discussion about the oxidized state of Ni based on the results of XPS and XRD. It says “The oxidized states of Ni were detected in the XPS spectra for both Pd-Ni and Pd-Ni-P samples; however, no NiO peaks (37.3 o, 43.3 o, 62.9o, and 87.0o of 2θ) were observed in XRD patterns (Figure 2a). This might also indicate the formation of Pd-Ni alloy and Pd-Ni-P alloy.” (Lines 205-208)

- The analysis of the XPS results needs to be improved. In the preparation of the catalysts, the authors do not indicate that the Pd-Ni-P/ catalyst is subjected to reduction treatment. The Pd-Ni-P/C and Pd-P/C catalysts were heat treated in inert (N2) up to 500°C.

 The Pd signal must be deconvoluted and the contributions of the oxidized and reduced species must be shown. This same analysis must also be carried out on the spectra of the catalysts for nickel.

Answer 8. Thank you for your question. The Pd signal was deconvoluted into Pd oxide and metallic Pd for both Pd-P and Pd-Ni-P (Figures S2a-S2b). The results show that although metallic Pd and Pd oxide coexist, metallic Pd dominates. Same analysis was also carried out on the spectra of the catalysts for Ni (Figures S2c-S2d). (Lines 189-202)

-Despite the low yields, the catalytic results are very interesting. However, they should be discussed again after knowing more precisely the nature of the supported phases.

Could acidity provide an activity-promoting effect?

 Answer 9. Thank you for your good question. Since all the samples use activated carbon as the carrier, the acidity of the catalyst itself is probably very low. In Figure 5c, we suggest that P in Pd-P or Pd-Ni-P catalyst plays a role on the adsorption of -OH group in lignin molecule and, together with Pd, facilitates hydrolysis of β-O-4 ether bond. From this viewpoint, the Pd-P and Pd-Ni-P catalysts might have a low acidity. (Lines 309-312)

Did the authors study the reusability of the more selective catalyst?

 Answer 10. Thank you for your question. The stability of the catalyst was not tested for lignin depolymerization, though the catalytic performance could be maintained for at least four cycles in the reaction of lignin model compounds. So far, it is difficult to recover the catalyst after the lignin depolymerization reaction. Our future experiments will focus on the easy recovery and reuse of the catalyst by optimizing the reaction process and equipment.

Reviewer 2 Report

In this manuscript, the authors present Pd-based nanocatalysts and the results of their application in the depolymerization of alkaline lignin. The physicochemical properties of the synthesized samples have been characterized in detail. The catalytic characteristics were investigated. The results obtained are significant for studying the catalytic conversion of lignin and directing the reaction pathways to the desired chemicals. Therefore, I recommend that the paper be published in the «Materials». However, the following issues should be considered before publication.

1.             In the abstract, the line 15 there are “we prepared four types of Pd-based nano-catalysts…”. But the authors synthesized three samples, and “The catalyst Pd/C was purchased from Beijing Hawk Science & Technology Co. Ltd”. I recommend writing correctly.

2.             Instruments and scanning conditions (XRD, XPS, TEM) should be added to the Materials and methods section.

3.             Formula 5, line 134. The selectivity is calculated relative to the masses of all products. How were the masses of the products determined? I think you should calculate by concentration or moles, not by masses. I also recommend clarifying the composition of the products.

4.             XRD analysis. How can we explain the absence of metallic nickel in Pd-Ni-P/C as opposed to Pd-Ni/C?

5.             Line 169, “It is suggested that the catalytic property of metallic Ni will play a role in this catalyst.” In the Discussion section “In ternary Pd-Ni-P catalyst systems, the existence of Ni enhances the water dissociation ability”. Why then did the Pd-Ni/C bimetallic sample displayed the lowest activity?

6.             I would recommend giving a suggested explanation for the synergistic effect and the role of P in Pd-Ni-P/C catalyst.

7.             Line 268, you need to replace the semicolon.

Author Response

Dear reviewer,

Thank you so much for your constructive questions and comments. We addressed all the questions and comments and the revised parts were highlighted with red color in the manuscript and supporting materials.

  1. In the abstract, the line 15 there are “we prepared four types of Pd-based nanocatalysts…”. But the authors synthesized three samples, and “The catalyst Pd/C was purchased from Beijing Hawk Science & Technology Co. Ltd”. I recommend writing correctly.

Answer 1: It was corrected in the revised manuscript. Please find it in the abstract part of the revised manuscript.

  1. Instruments and scanning conditions (XRD, XPS, TEM) should be added to the Materials and methods section.

Answer 2: Instruments and scanning conditions (XRD, XPS, TEM) have been added to the Materials and methods section. Please refer to Lines 138-144.

  1. Formula 5, line 134. The selectivity is calculated relative to the masses of all products. How were the masses of the products determined? I think you should calculate by concentration or moles, not by masses. I also recommend clarifying the composition of the products.

Answer 3: Thank you for your constructive question. The determination of product masses was carried out by calculating the concentration of the liquid product using the GC-MS standard curve, as described in Equations (1), (2), and (3) on page 3. The mass of each product was then calculated based on the total volume of the liquid product obtained. Since the total volume of liquid products remains constant, the selectivity can be calculated using their concentrations. For further clarity, equations (4) and (5) have been modified accordingly (please refer to Lines 134-136, the revised manuscript).

  It is worth noting that the products resulting from lignin depolymerization are known to be complex, making it challenging to calculate the moles of each individual product and the total moles of products. Therefore, the calculation of selectivity based on moles was difficult in this study. Furthermore, the composition of the products obtained from lignin depolymerization has been identified using GC-MS analysis and by referencing relevant literature reports. Given the wide range of reaction products, expressing selectivity based on mass might be the most suitable approach to demonstrate the catalyst's preference for certain products.

  1. XRD analysis. How can we explain the absence of metallic nickel in Pd-Ni-P/C as opposed to Pd-Ni/C?

Answer 4: Thank you for your constructive question. In our opinion, XRD analysis could just tell the absence of metallic Ni in Pd-Ni-P/C as opposed to Pd-Ni/C. Alloy formed in both catalysts. With XPS, we could observe the absence of metallic Ni in Pd-Ni-P/C as opposed to Pd-Ni/C. This might be due to the different fabrication methods of the two catalysts. Pd-Ni/C was synthesized by an impregnation method. The material was annealed under H2 atmosphere during the procedure. Therefore, there is metallic Ni observed. In contrast, Pd-Ni-P NPs were prepared with oleylamine (OLA) as the reducing reagent and annealed under inert gas.

  1. Line 169, “It is suggested that the catalytic property of metallic Ni will play a role in this catalyst.” In the Discussion section “In ternary Pd-Ni-P catalyst systems, the existence of Ni enhances the water dissociation ability”. Why then did the Pd-Ni/C bimetallic sample displayed the lowest activity?

Answer 5: Thank you for your constructive question. For Pd-P catalyst, due to the existence of Pd-P bonds, water is firstly adsorbed and dissociated, and the catalyst can selectively generate alkene intermediates as shown in Figure 5c. For Pd-Ni-P catalyst, the addition of Ni could further improve the dissociation of water and adsorb more OH*, which could weaken the hydrodeoxygenation ability of the catalyst to produce more guaiacols.

  Pd-Ni/C contains rich metallic Ni, it is suggested that it exhibits stronger hydrocracking and hydrogen transfer capabilities, resulting in more aromatic ring hydrogenation products (i.e. aliphatic products, Figure 3 and Table 2). As a result, Pd-Ni/C might expend more effort in hydrogenating monomers but less in depolymerizing lignin, which could lead to lower yields of depolymerization products.

  1. I would recommend giving a suggested explanation for the synergistic effect and the role of P in Pd-Ni-P/C catalyst.

Answer 6: Thank you for your constructive question. A suggested explanation has been added for the synergistic effect and the role of P in Pd-Ni-P/C catalyst in the revised manuscript. It says “Herein, it is suggested that P could form coordinating bonds with -OH groups of lignin molecules and, together with Pd, facilitates hydrolysis of β-O-4 ether bond. In another word, P in Pd-P or Pd-Ni-P catalyst might play a role on the adsorption -OH group of lignin, and from this viewpoint, the two catalysts might have a low acidity.” Please refer to Lines 309-312.

  1. Line 268, you need to replace the semicolon.

Answer 7: Thank you for your consideration. It has been revised. Please refer to Line 306.

Reviewer 3 Report

In this paper the authors describe a ternary catalytic system that is capable of producing guaiacols with high selectivity. The system has been previously studied by the authors for lignin derivatives. I find the research of interest and worth publication. My biggest critique of this paper is that the yield of guaiacols is relatively low from a commercialization perspective. It would be helpful if the authors compare guaiacols yields for other comparitive catalysts to show improvement over other commercial catalysts.

Author Response

Dear reviewer,

Thank you so much for your constructive comments. We addressed all your comments and the revised parts were highlighted with red color in the manuscript and supporting materials.

In this paper the authors describe a ternary catalytic system that is capable of producing guaiacols with high selectivity. The system has been previously studied by the authors for lignin derivatives. I find the research of interest and worth publication. My biggest critique of this paper is that the yield of guaiacols is relatively low from a commercialization perspective. It would be helpful if the authors compare guaiacols yields for other comparitive catalysts to show improvement over other commercial catalysts.

Answer: Thank you for your constructive comment. For comparison, the product information of lignin depolymerization promoted by various previously reported metal-based catalysts was illustrated in Figure S3. The related statement was also added in the section of Conclusions (Line 327, Page 10).